# Microhabitat Conditions and Inter-Species Competition Predict the Successful Restoration of Declining Relict Species Populations

**DOI:** 10.3390/ijerph20010608

**Published:** 2022-12-29

**Authors:** Andrzej K. Kamocki, Aleksander Kołos, Magdalena Pogorzelec, Małgorzata Ożgo

**Affiliations:** 1Faculty of Civil Engineering and Environmental Sciences, Bialystok University of Technology, 15-351 Białystok, Poland; 2Department of Forest Environment, Białystok University of Technology, 15-351 Białystok, Poland; 3Department of Hydrobiology and Protection of Ecosystems, University of Life Sciences in Lublin, 20-626 Lublin, Poland; 4Department of Evolutionary Biology, Kazimierz Wielki University, 85-093 Bydgoszcz, Poland

**Keywords:** glacial relicts, Knyszyn Forest, reintroduction, *Salix lapponum*, wetland

## Abstract

The local populations of relict plant taxa living near the limits of their geographical range are particularly vulnerable to extinction. For example, *Salix lapponum* is one of the rarest and most endangered glacial relicts in Western and Central Europe. In Poland, the number of its sites has dramatically decreased over the past few decades, prompting us to take conservation measures focused on saving endangered populations. During a field experiment aimed at the reconstruction of the downy willow population in the Knyszyn Forest (NE Poland), 730 individuals of the species were planted in four different natural sites. The seedlings were obtained by micropropagation from parts of vegetative individuals taken from the most abundant population of this species in eastern Poland (Lake Bikcze). The success of the reintroduction, measured by the number of individuals that survived 2–3 years in the wild and took up growth, was about 67%, however, with low flowering efficiency (7.5%). Additionally, monitoring showed significant differences in plant survival and growth rates under different habitat conditions prevailing at the site and with different cover from competing species, especially tall grasses. However, the restoration projects for relict shrub species should include periodic removal of competing plants and protection of plants from trampling and browsing by herbivorous mammals to increase reintroduction success.

## 1. Introduction

In the face of intensifying global climate change and the associated disruption of local habitat and biocenotic conditions [1]), numerous plant species are experiencing many negative consequences. Particularly vulnerable are boreal taxa whose populations grow near the limits of their geographical range [2,3,4]. Their occurrence in Central and Western Europe is currently limited to scattered sites in the mountains or on peatlands in the lowlands [5]. Further, relict species are disappearing mainly due to global climate warming and the associated groundwater deficit, as well as increased secondary succession caused by the cessation of agricultural use of wetlands [6,7]. In the absence of management, wetlands can be subject to vegetative succession that leads toward a recovery of shrub or forest communities dominated by *Betula* spp., *Salix* spp., or *Alnus glutinosa* [8,9,10,11,12]. Additionally, these processes contribute to population decline in the number of relict species and usually cause their extinction [13], as these plants are mostly heliophytes and generally lose in competition with expansive woody species [14,15]. The relict taxa are often found in isolated sites with heavily transformed habitats. The scarce availability of suitable niches is the reason for significant limitations in the efficiency of reproduction and seed dispersal. Further, the local populations of glacial relict taxa living in such conditions at the edges of geographic ranges are particularly vulnerable to extinction [16,17,18,19,20].

One of the most endangered glacial relic plant species found in isolated sites in Western and Central Europe is *Salix lapponum* (downy willow). It is protected and listed in “red books” in many countries, including Lithuania, Poland, Ukraine, and the UK [21,22,23,24]. However, the species is abundant only in Scandinavia and northwestern Russia [25]. In the Baltic countries and Eastern Europe (Estonia, Latvia, Lithuania, Belarus, and Ukraine), it occurs much less frequently, only in scattered localities [26,27,28]. In Poland, *Salix lapponum* belongs to the rarest components of the native flora and occupies sparse biotopes in the Sudetes and in the eastern and northeastern parts of the country [21,29]. Notably, the species prefers acidic peat soils with pH 4.0–4.5, or peat-silty soils with pH 5.0–7.0 [17,20,30]. The *S. lapponum* grows in mesotrophic fens and transitional mires as well as in wet meadows (*Molinietalia* alliance). The species also appears in thickets with *Betula humilis*. In the Karkonosze Mountains, downy willow forms the community *Salicetum lapponum* [31]. Additionally, in recent decades, many local populations of *S. lapponum* in the Polish lowlands have become extinct or have experienced a dramatic decline in population size [30,32,33,34]. For example, in northeastern Poland, the species can be found at several sites in the Lithuanian Lake District, in the Biebrza valley, and in the Białowieża and Knyszyn Forests. However, only extremely small populations have survived in all lowland sites. Due to increasing environmental threats, their persistence in Poland is uncertain. The decline in the population of *S. lapponum* is mainly caused by changes in habitat conditions in natural sites and their isolation. A serious potential threat to the species is also the possibility of creating interspecific hybrids, which are the result of crossing downy willow with other species of the genus *Salix,* which is common within this group of plants [35,36,37].

The unfavorable environmental changes in recent decades have made saving endangered species a priority not only for decision-making bodies in many countries but also for international organizations. For example, there are legislative initiatives aimed at preventing the extinction of endangered species that have appeared both at the European level (e.g., Habitat Directive 92/43/EEC, the EU’s biodiversity strategy for 2030) and at the global level (e.g., Convention on Biological Diversity). The reintroduction of species is becoming one of the most important strategies for wildlife conservation worldwide [38,39,40,41]. Additionally, the restoration of populations that have become extinct (or augmentation of existing populations) is the essence and primary goal of reintroduction [42]. However, because generative reproduction in many plant species can be limited by a lack or scarcity of propagules [43,44] as well as inbreeding, reintroduction is often the only way to preserve endangered species in disturbed habitats, especially for extremely small populations [45].

The restoration of declining plant species to areas previously occupied by them is a fairly common active conservation measure in many countries around the world [46,47]. For example, in Europe, restoration programs involving habitats with rare species of the genus *Salix* are carried out in the United Kingdom [48], the Czech Republic [49], and Poland [50], among others. These programs are usually linked to the verification of methods that increase the effectiveness of conservation treatments and to multifaceted studies of factors affecting the stability of populations of endangered species [15,16,17,20,34,51,52,53,54,55,56,57,58,59].

In the years 2017–2020, a program of reconstruction of the *Salix lapponum* population was carried out in eastern Poland [50]. The plants were obtained by the method of micro-propagation from parts of vegetative individuals from natural populations in the Łęczna-Włodawa Lakeland. Additionally, prepared and acclimatized plants were introduced in 2018–2020 to selected peatland habitats, including locations in the Knyszyn Forest. Furthermore, to assess the success of species conservation activities, a number of studies and observations were carried out to answer the following questions: What is the survival rate of translocated plants and their fitness at their new sites? Is the success in reintroduction efforts influenced by abiotic or biocenotic environmental factors in the new habitats?

## 2. Materials and Methods

### 2.1. Study Site

The study was conducted on two physiographic sites: the Stare Biele mire, located in the central part of the Knyszyn Forest, and the quaking bog in the vicinity of Lake Wiejki (NE Poland) (Figure 1).

Lake Wiejki is located in the upper reaches of the Supraśl River, in the right bank part of the valley, within a broad peat-filled extrusion basin (Gródek-Michałowo Depression), the relief of which was shaped during the Riss glaciation [60]. The area is flat, with few elevations and hills lying on the periphery of the basin. Although the wetlands surrounding Lake Wiejki have been significantly transformed following land reclamation work that began in the late 1950s, the zone adjacent to the lake still retains well-watered mires. A zonal arrangement of aquatic and marsh vegetation is characteristic of the lake. Until the beginning of the past decade, macrophytes (*Nymphaea alba* and *Potamogeton natans*) were abundant in the littoral zone of the lake. Currently, following the shallowing of the lake, there are clear symptoms of the regression of macrophytes, which are being displaced by *Equisetum fluvialitle* in the litoral zone and by *Ceratophylletum submersi* in the limnetic zone. Further, the zone of quaking bog is mainly occupied by peat-sedge communities dominated by *Carex rostrata* and mosses belonging to the genus *Sphagnum*. The complex of open communities is surrounded by thickets with *Salix cinerea* and forests dominated by *Betula pubescens, Pinus sylvestris,* and *Alnus glutinosa*. The once common shrub communities with *Salix lapponum* are no longer present here. Additionally, the last stand of this species in the vicinity of Lake Wiejki disappeared in the mid-1990s [61]. Since 2005, the lake and the surrounding peatland have been under the protection of a nature reserve.

A local population of *Salix lapponum* has not existed at Lake Wiejki for more than 25 years. The sections of the lakeside peatland selected for *S. lapponum* reintroduction were well-watered, sunny, and not threatened by tree and shrub invasion in the near future. The planting of individuals was planned in the northern and eastern parts of the peatland within the zone of quaking bog, in an area covered by plant communities including *Carex rostrata*, *Menyanthes trifoliata*, *Comarum palustre,* and *Sphagnum ssp*. Furthermore, there were patches in the eastern part of the site with high coverage of *Equisetum fluviatile* and *Calamagrostis stricta*. Selected fragments of the wetland were open enclaves surrounded by forest and shrub communities and were only rarely covered with single specimens of *Betula pubescens*.

Stare Biele is a large complex of wetlands, surrounded by moraines and kame hills, that developed in the Riss glaciation [62]. It is characterized by natural forest and swamp communities with numerous rare and protected plant species [63], covering a total area of about 300 hectares. Among the most common habitats are bog-alder forests (*Ribeso nigri–Alnetum* and *Sphagno squarrosi–Alnetum*), bog-birch forests (*Thelypterido–Betuletum pubescentis*), and bog-spruce forests (*Sphagno girgensohnii–Piceetum*). In addition, in the northeastern and eastern parts of the wetland, large areas are covered by semi-natural sedge meadows. The largest area is occupied by a *Carex elata* community with a *Salix lapponum* stand. This is the only site of this relict plant in the Knyszyn Forest [64]. The maps from the early nineteenth century and analyses of the upper layer of the peat deposit indicate that in these parts of the peatland, open wetlands have dominated for centuries [65]. The sedge meadows were mowed mainly during dry periods until the early 1970s. Currently they are occupied by expansive trees and shrubs: *Betula pendula, B. pubescens, Pinus sylvestris, Picea abies, Salix cinerea,* and *S. pentandra* [10]. Additionally, a stream named Derazina flows out of the peatland. The complex of wetlands, as well as the direct catchment area of the Stare Biele site, is neither regulated nor embanked, and the Derazina River valley is characterized by fairly natural hydrological patterns. The valley is mainly covered with tall sedge communities. Small fragments of fens with a dominance of low sedges (*Carex lepidocarpa, C. panicea,* and *C. nigra*) are still preserved in the near-edge zone of the valley. In 1987, a significant part of the peatland, together with stands of downy willow, was included in the nature reserve.

Further, the status of the *Salix lapponum* population located in the Stare Biele mire has been monitored since 1994 [15]. The studies have shown that the local population of downy willow is stable but strongly threatened with extinction, mainly due to its very low abundance. The, opportunities to increase the population’s range are limited due to secondary succession and competition from trees and shrubs. It was assessed that the most effective and fastest way to strengthen the population of *Salix lapponum* in the peatland is the introduction of new individuals (population strengthening).

### 2.2. Reintroduction Experiment

The plants obtained through micro-propagation in tissue cultures were used to restore the *Salix lapponum* population (using methods developed by Pogorzelec et al. [50,66]). The plant material (over 300 individuals) came from the largest downy willow population in eastern Poland, growing in a peat bog by Lake Bikcze (N 51°22.724′ E 23°02.563′). This population was chosen because it had a satisfactory sex ratio (3♀:1♂), high genetic variation, and no observed clonal individuals [55].

Additionally, after reaching an appropriate size (the main shoot was about 25 cm long), plants were transferred from in vitro cultivation to a specially prepared substrate mimicking the edaphic conditions of peatland habitats and kept at a controlled temperature and humidity for about 3 months. Subsequently, *Salix lapponum* rooted individuals were transferred to specially prepared cold frames located near Poleski National Park (eastern Poland) to acclimate in conditions similar to those prevailing at natural sites. However, after 4 weeks to 6 months, the plants were transported to the sites where their introduction to the environment was planned. The specimens intended for planting were woody and had several side shoots, the formation of which was induced by repeated cutting of the main shoots during the acclimation term [50].

In all selected locations, *Salix lapponum* was planted in aggregations of irregular shapes, and the plots ranged in size from about 1.5 to 4 m^2^. The individuals were planted with the entire root ball directly into the quaking bog at a spacing of 20–30 cm. Finally, all the actions were carried out in consecutive years (2018–2020) in the periods from July to October. 

In the eastern part of the Lake Wiejki nature reserve (site WE), 80 individuals of *Salix lapponum* were planted in three clusters (15, 30 and 35 individuals) in 2018. In 2019, additional 120 plants were introduced there in three new clusters (2 × 24 and 1 × 72). During the same growing season, 240 individuals (4 × 48 and 2 × 24) were introduced into the vegetation patches in the northern part of the lake (site WN). In autumn 2020, 3–4 male specimens were introduced within selected *Salix lapponum* plantings (26 specimens in total). 

In 2018–2020, as a result of reintroduction activities carried out at Lake Wiejki, a new population of 466 specimens of *Salix lapponum* was established.

In September 2019, preliminary planting of *Salix lapponum* in the Stare Biele nature reserve was carried out in two locations: in the direct neighborhood of existing stands of *Salix lapponum* in the Stare Biele mire (site SB) and at the edge of the Derazina River valley (site DD). In the first location, 216 individuals were planted in 9 clusters of 24 individuals each. The plantings were carried out in plant patches with a dominance of *Carex elata* and significant cover of *Carex appropinquata*, *Carex lasiocarpa,* and *Menyanthes trifoliata*, in which competing tree and shrub species occurred sporadically (cover up to 30%). Additionally, in the vegetation patches, there were sparse young specimens of *Betula pubescens* and *Salix cinerea*, as well as *Alnus glutinosa*, *Picea abies*, *Salix pentandra*, *S. rosmarinifolia,* and *S. myrsinifolia*. In the marginal part of the Derazina River valley, about 1 km east of the existing *Salix lapponum* site, a patch of well-preserved sedge-moss percolation mire with *Carex panicea* and *Carex lepidocarpa* was selected as an alternative habitat, within which 96 individuals were planted in three aggregations (2 × 24 and 1 × 48 individuals) in 2019. As part of the strengthening of the existing population of *Salix lapponum*, a total of 312 specimens were planted in the Stare Biele mire in 2019. 

In 2022, four years after the beginning of the experiment, the number of surviving plants, the number of flowering individuals, and the height of above ground shoots were determined within each *Salix lapponum* aggregation. In addition, the cover of dominant and competing herbaceous species as well as trees and shrubs was determined, and the height of the herbaceous layer was measured (20 measurements). In three aggregations in the eastern part (a total of 120 individuals) and in one cluster in the northern part of the lake (48 individuals), the height of the plants was measured before planting. The measurements were repeated in the field in the summer of 2022 to determine the growth rate of aboveground shoots of *Salix lapponum*.

### 2.3. Groundwater Quality

The samples of the shallow groundwater/peat soil solution from the root zone (0–20 cm) were collected from the introduction sites with approximately monthly frequency during the optimum growing season (May–September 2022; *n* = 5). The samples were collected in 500 mL PE containers and transported to the laboratory for spectrophotometry and chromatography analyses of the concentration of biogenic substance fluxes: nitrogen fractions total nitrogen (N_Tot_; UV-1800 spectrophotometer Shimadzu, Japan), dissolved inorganic nitrogen (DIN), ammonia nitrogen N-NH_4_^+^, nitrates N-NO_3_^−^, nitrites N-NO_2_^−^, phosphorus fractions-total phosphorus (P_Tot_), phosphates (P-PO_4_^−3^), sulphates (S-SO_4_^2−^), and total organic carbon (TOC) was determined in TOC-L analyser with SSM-5000A Solid Sample Combustion Unit (Shimadzu, Japan). The ion chromatography method was used to determine the hardness of the water and the concentrations of ions—Na^+^, K^+^, Ca^2+^, Mg^2+^, Li^+^, Cl^−^, F^−^ (Dionex ICS-1100, GMI, New York, NY, USA). In addition, the specific conductance (EC) and reaction (pH) were determined directly in the field using a Hach HQ40D measuring device (Hach Lange, Loveland, CO, USA).

### 2.4. Data Analysis

The significance of differences in the height of *Salix lapponum* shoots in different habitat conditions and in consecutive years of the experiment was tested by one-way analysis of variance (ANOVA) and the Kruskal-Wallis test (variables were not normally distributed). Further, the normal distribution of the raw data was tested with the Shapiro-Wilk test at the significance level of *p* < 0.05. The homogeneity of the variance of the samples was examined by Levene’s test. In addition, statistical calculations were made in the JASP 0.16.2 program (JASP Team, Amsterdam, The Netherlands, 2022). The interrelationships between the survival rates of *Salix lapponum* individuals, acrotelm water chemistry, and competition from other vascular plants (the cover and height of the herbaceous layer) were analyzed by principal component analysis (PCA) with XLStat (Addinsoft, Paris, France).

## 3. Results

### 3.1. Changes in the Number of Individuals

During the experiment of *Salix lapponum* population reconstruction implemented in 2018–2020, a total of 730 individuals of the species were planted. The success of the reintroduction of downy willow in the Knyszyn Forest in 2022, measured by the number of individuals that survived in the wild and took up growth, was about 67% (492 introduced specimens survived more than two years after the completion of the plantings). The highest reintroduction success rate was recorded at site DD (85.42%), with 100% of live individuals found in one of the aggregations (Table 1). A slightly lower survival rate of downy willow (81.55%) was recorded at site WN. Further, between 68% and 94% of previously planted individuals have survived here in individual aggregations. The survival was markedly lower at the other two sites: WE–57.55%, SB–55.56%. In one of the aggregations at site WE, no planted individuals survived, while at site SB, in four of the nine aggregations, reintroduction success was less than 50% (Table 1).

### 3.2. Changes in Shoot Height

The *Salix lapponum* individuals measured two or three years after the introduction into the natural environment reached between 9 and 79 cm in height (average 38.19 ± 14.17). The tallest *Salix lapponum* shoots were recorded in 2022 at site WE (average 46.01 ± 17.28, maximum 79 cm), while the shortest shoots were observed at site DD (average 26.73 ± 7.99, maximum 46 cm). However, both of these values are significantly different with respect to the values recorded at sites SB and WN (*p* < 0.001; Kruskall-Wallis test; Figure 2), while the aboveground shoots reached intermediate sizes (37.61 ± 11.5 and 38.51 ± 11.8, with a maximum of 69 cm and 70 cm, respectively), and the differences between them were not statistically significant.

There were no significant differences in the distribution of the studied parameter’s values for the SB and WN sites—most values were in the middle ranges of the scale, 31–40 cm and 41–50 cm (Figure 3). However, significant deviations from this pattern were noted at site DD—the individuals growing there were significantly shorter, and more than 50% reached only 21–30 cm in height. The fractions of the tallest individuals were the most numerous at site WE–only here there were specimens with a height exceeding 70 cm.

The monitoring of the reintroduced individuals of *Salix lapponum* within selected aggregations in the vicinity of Lake Wiejki indicated significant differences in the growth rates under different habitat conditions prevailing at the sites. The measurements taken just before the individuals were planted in 2019 at the WE and WN sites showed no significant differences in their height (17.31 ± 4.73 and 17.65 ± 5.23, respectively). Tests conducted three years later (2022) showed that individuals grew significantly faster at site WE and were almost twice as tall as at site WN (55.08 ± 15.89 and 34.08 ± 9.46, respectively; Mann-Whitney U test, *p* < 0.001; Figure 4).

### 3.3. Flowering Efficiency

The single-flowering individuals were observed as early as the first year after they were planted. For example, in 2022, flowering individuals of *Salix lapponum* were observed only at sites located in the vicinity of Lake Wiejki, where only female plants developed inflorescences. However, the proportion of flowering individuals in the population was much greater in the case of site WN–27, where out of 168 shoots that survived (16.07%), had developed inflorescences (Table 1). In contrast, at site WE, flowering was undertaken only by 10 out of the 122 individuals recorded there (8.2%). At sites SB and DD, no inflorescences were recorded on any of the *Salix lapponum* individuals.

### 3.4. Interrelations between the Survival Rates of Salix lapponum, Acrotelm Water Chemistry, and Competition from Other Vascular Plants

The PCA of the interrelation between the survival rates of *Salix lapponum*, acrotelm water chemistry, and competition from other vascular plants showed a total loading on the first two axes of 78.9% (Figure 5, Table 2).

The survival rates of *Salix lapponum* were negatively correlated with the cover and height of the herbaceous layer but did not show significant correlations with the measured water chemistry parameters. This contrasts with the cover and height of the herbaceous layer, which showed correlations with a number of water chemistry parameters (Table 2).

## 4. Discussion

### 4.1. The Effects of Reintroduction

Each reintroduction project, because of the species covered and the circumstances of its implementation, should be considered a separate and unique case. However, there are a number of elements linking actions for restoring extinct species or enhancing declining populations that influence the course and final results of reintroduction [42,67,68,69]. These include both biological factors (e.g., type and quality of propagules, the number and location of source populations, habitat features, competition from invasive species) and methodological factors (e.g., techniques for preparing and planting propagules, site preparation, and timing). However, a review by Godefroid et al. [46] shows that only three factors are significant for the success of reintroduction: the origin of the plant material, the removal of potential competing plants in sites of endangered species reintroduction, and the location of these sites in protected areas, while it may be limited by factors such as inadequate habitat selection, habitat degradation, misidentification of the causes of population decline, insufficient numbers of reintroduced individuals, and a short project duration.

The survival rate of individuals, the ability of reintroduced individuals to bloom and produce fruit, and the recruitment of plants are usually considered measures of reintroduction success. These factors determine the ability of populations to form new generations [42,46]. As part of the ongoing *Salix lapponum* reintroduction project in the Knyszyn Forest, a total of 67.4% of previously introduced individuals survived two to three years after the completion of planting. As a result, the result ranks well above the average survival rate of 52%, estimated on the basis of the effects of several hundred population restoration projects of endangered plant species implemented so far in the world [46]. For *Salix lapponum*, the survival rate of reintroduced individuals was reported to be 19%–52% (after 7 years; [70]) and 43–100% (after one year; [50]). The high efficiency of our experiment may be due to both the location of the sites within protected sites and the use of individuals prepared on the basis of material taken from the population of *Salix lapponum,* characterized by a large intra-population diversity that has existed in eastern Poland for a long time [55].

The relatively high survival rate of the specimens planted in 2018–2020 may also be due to the short duration of the experiment. Some studies indicate that in the case of *Salix lapponum*, a preliminary assessment of the species’ restoration effects is possible after just one year [50]. However, Mardon [70] believes that conservation projects for endangered willow species should be planned for decades rather than years. Additionally, the high survival rate of individuals in the early years of the experiment may decrease dramatically in subsequent periods. Drayton & Primack [71] noted the disappearance of almost all populations of perennial plant species that had been reintroduced in nature reserves 15 years earlier.

The high efficiency of our experiment can also be linked to the method used to implement individuals into plant patches by planting them in dense aggregations. Such structures may have favored the survival of plants in competition for limited resources. According to Bertoncello et al. [72], some woody species have higher survival rates when growing in aggregations. This trait can be exploited when introducing nonpioneer species into disturbed habitats.

Possibly, the large number of reintroduced individuals (730) and the way in which they were prepared also influenced the apparent success of the species’ restoration in the Knyszyn Forest. Godefroid et al. [46] emphasize that in the reintroduction projects of endangered plant species performed so far, an average of 400 individuals were introduced, while the survival rate of seedlings is significantly higher when the number of propagules approaches 1000.

The success of reintroduction may also be influenced by factors related to the preparation of the plants. First, the maternal population should be numerous and genetically diverse [42]. The second decisive factor is how the plant material is prepared. If it is not possible to obtain it from seeds, an alternative method is to use tissue cultures. The process of micro-propagation should be adapted to the specifics of the species, and the obtained specimens should be acclimated before being transferred to the environment [50,73].

The downy willow plants introduced to the peat bogs of the Knyszyn Forest were created in vitro from individuals from the most abundant population of this species in eastern Poland, growing in the bog at Lake Bikcze (Łęczyńsko-Włodawskie Lakeland). The methods to obtain a large number of plants (various clones) have been established in over 20 years of experience. The two-stage acclimatization of plants, carried out in the laboratory and then in a special field station, increased the survival potential of the plants in the harsh conditions of their natural habitats [73].

The findings that the use of pot-rooted plants results in lower survival rates in the early years of reintroduction [46] were not confirmed in our study. Furthermore, all plants of *S. lapponum* were grown ex situ in pots with a substrate that imitated soil conditions in the target reintroduction habitats before they were transferred to the environment.

The flowering and fruiting rates in reintroduction projects are generally low (16–19%) and tend to decline over time [46]. In addition, most of the reports on the restoration of endangered plant species include information on the absence or sporadic recruitment of plants. Further, observations made during our experiment confirm these reports. The percentage of blooming individuals of *Salix lapponum* was only 7.52% after 2–4 years, and only female individuals planted in the vicinity of Lake Wiejki formed inflorescences. No new specimens of downy willow were found at reintroduced sites. Pollination is closely related to the short distance between male and female individuals [74], and only female individuals of *Salix lapponum* generally produce numerous seeds if male individuals grow at a distance of up to 5 m [70]. However, since both female and male individuals were planted in clusters in our project, it is possible that individuals will appear in the future, especially if suitable germination sites are present, such as bare ground created by animal impact [75].

In our view, the survival rate of introduced plants and their fitness may also be significantly affected by trampling and browsing by herbivorous mammals (especially moose, which are abundant in the wetlands of the Knyszyn Forest), with some plants being destroyed by animals. In Scandinavia, reindeer readily feed on downy willow, and the effects of browsing are more pronounced on young plants [76,77,78]. Additionally, the authors mentioned that reindeer browsing significantly reduced (by 30–50%) the number and height of individuals.

### 4.2. Effect of Competitor Abundance on Reintroduction Success and Fitness of Introduced Plants

The overall result of *Salix lapponum* reintroduction in the Knyszyn Forest, measured by the number of survivors, indicates a high effectiveness of this conservation method, although the survival rate of individuals in particular clusters was highly variable. Additionally, only in one of the groups did all the planted individuals survive. In seven of the 22 aggregations, the average proportion of living individuals was less than 50%. The low survival rate of plants in some locations can be linked primarily to the presence of competing species in the phytocoenoses. Due to their type of growth strategy and life form, grasses, sedges, and tall megaforbs are able to effectively inhibit the growth of other plants. The species with an iterative growth type are particularly expansive in wetlands, especially under conditions of disturbed water relations [79,80]. The downy willow planted in patches dominated by *Calamagrostis stricta* (site WE) was particularly vulnerable to the inhibitory effects of competing species. In one of the clusters, all of the introduced individuals disappeared, while in the other, only eight of the 24 planted individuals survived. The competitiveness of *Calamagrostis stricta* is mainly due to its ability to form a dense layer, tightly covering the ground surface. This grass, which grows up to almost 2 m, restricts the living space of plants of smaller size, strongly shades the ground, and competes with small *Salix* individuals, which usually measure 10–20 cm. Interestingly, we noted no negative impact on plant survival from the equally expansive common reed, *Phragmites australis*. Within the boundaries of the transition mire located at the edge of the Derazina River valley (site DD), this species is one of the dominants, but it reaches a short height (above-ground shoots of about 1 m in height) and a cover of less than 50%, which does not pose a major threat to downy willow individuals.

The survival rate of individuals planted in patches dominated by *Carex elata* was significantly lower (<50% in four aggregations out of nine). Although *Salix lapponum* specimens were planted in the empty spaces between sedge clumps, the canopy formed by their leaves shaded the sites and caused the death of young downy willow individuals. It was observed that introduced plant survival was noticeably lower in patches dominated by two co-occurring species: *Carex elata* and *Carex lasiocarpa*, while there was no negative effect of *Carex rostrata* (site WN and part of site WE). Due to sedge’s short aboveground stems, delicate leaves, and low cover, it did not pose serious competition to downy willow. In patches with this species, the survival rate of the studied plants was the highest (69–97%).

### 4.3. Effect of Habitat Conditions on Reintroduction Success and Individual Growth and Fitness

As a general rule, groundwater is essential for the formation of vegetation, biomass productivity, the decomposition of organic materials, and the physiographic structuring of the peatland surface. The PCA analyses confirmed that some habitat parameters could be considered conducive to the survival of reintroduced individuals of *Salix lapponum* in different types of habitat, but these relationships were not statistically significant. However, moderate or low electrical conductivity and low concentrations of total nitrogen, dissolved inorganic nitrogen, ammonia nitrogen, nitrites, total organic carbon, and lithium can be considered as favoring proper functioning and further growth of *Salix lapponum* individuals (marginally significant at *p* < 0.10). Certainly, the lower abundance of nutrients in groundwater influenced the reduction of competition from other plant species and the preferential conditions for the development of young individuals. The PCA results revealed that the cover of dominant and competing herbaceous species and the height of the herbaceous layer were the two factors that had a significant effect (*p* < 0.05) on the survival rate of reintroduced individuals.

## 5. Conclusions

The success of *Salix lapponum* reintroduction is significantly influenced by a number of factors, the most important of which are the method of seedling preparation, the number of individuals introduced into the wild, the presence of competing species in the vicinity of the newly created sites, and the type of habitat. For example, downy willow clearly prefers peatlands that are not very rich in biogenes—the survival rate of individuals introduced into nutrient-poor habitats was significantly higher than the rate recorded after willow reintroduction in more fertile habitats. In our opinion, it is necessary to remove competing trees and shrubs to protect introduced specimens of *Salix lapponum*, as this is one of the most important treatments that need to be undertaken for the survival of this shrub species in peatlands [15,54] and increasing important community metrics among grassland flora and fauna [81]. Additionally, reintroduction of downy willow into plant communities dominated by competing herbaceous species, especially grasses, which inhibit or even prevent the growth of introduced plants, should be avoided. Furthermore, the moss communities with negligible herbaceous plant cover seem to be the most suitable for such treatments. Finally, planting individuals in small aggregations of a few dozen specimens may favorably influence the success of the reintroduction of *Salix lapponum* in the initial years after the treatment, as the clustered population structure may favor the survival of individuals in competition for limited environmental resources.

## Figures and Tables

**Figure 1 ijerph-20-00608-f001:**
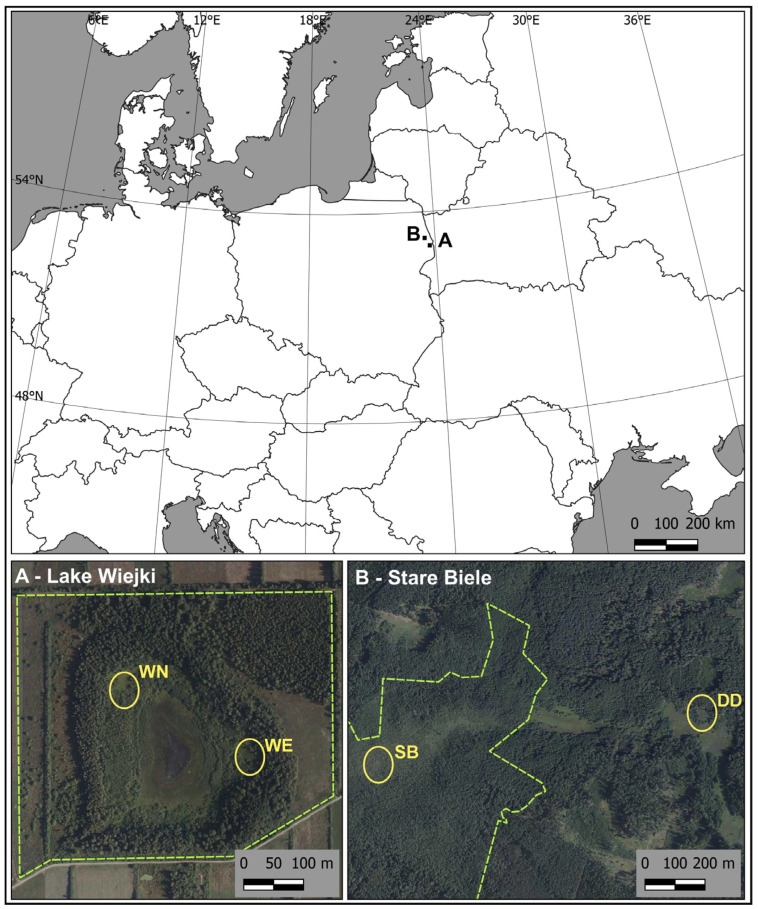
Map of the study area and location of reintroduction plots of downy willow. The yellow ellipses indicate an approximate location of reintroduction sites (WN, WE, SB, DD). The light green dashed line is the borderline of the nature reserves. The map was generated using the QGIS 3.16.4-Hannover software (Free and Open Source Software (FOSS); Free Software Foundation, Inc., Boston, MA, USA; www.qgis.org, accessed on 30 November 2022). Source for the orothophotomap: goeportal.gov.pl (Terms and conditions: https://www.geoportal.gov.pl/regulamin, accessed on 30 November 2022).

**Figure 2 ijerph-20-00608-f002:**
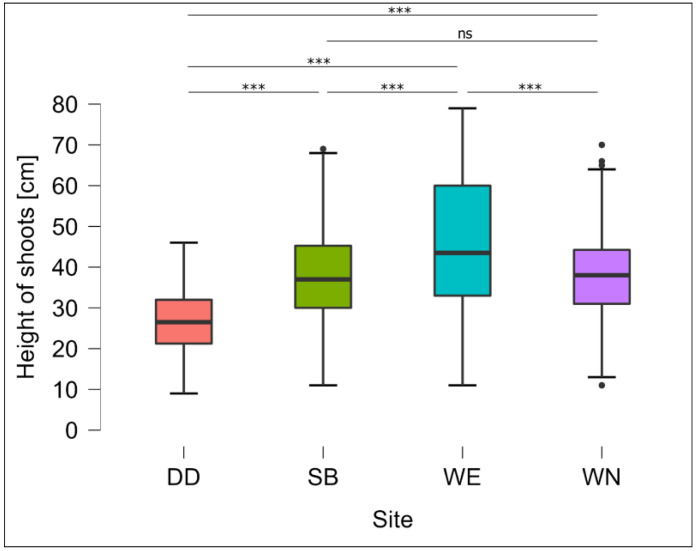
Variation in the height of *Salix lapponum* individuals two/three years after the completion of seedling planting on different types of habitats. SB—Stare Biele nature reserve, sedge-moss mire; DD—Derazina valley, sedge-moos percolation mire; WN—Wiejki Lake nature reserve north, active quaking bog; WE—Wiejki Lake nature reserve east, inactive quaking bog; ***—very significant (*p* < 0.001); ns—not significant (*p* ≥ 0.05).

**Figure 3 ijerph-20-00608-f003:**
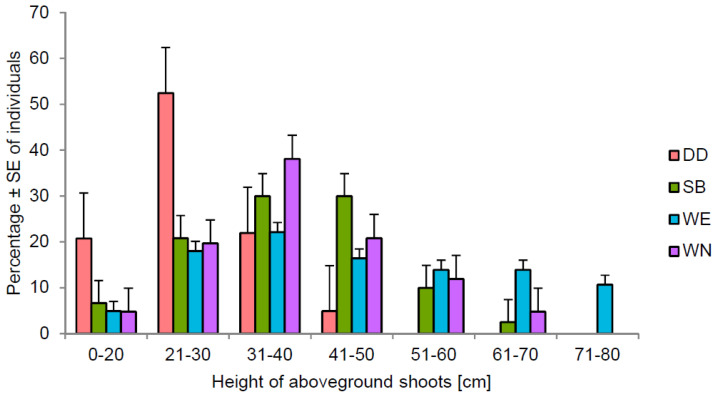
Size structure of *Salix lapponum* individuals two/three years after the reintroduction treatments of the species on different types of habitats. For explanations, see Figure 2.

**Figure 4 ijerph-20-00608-f004:**
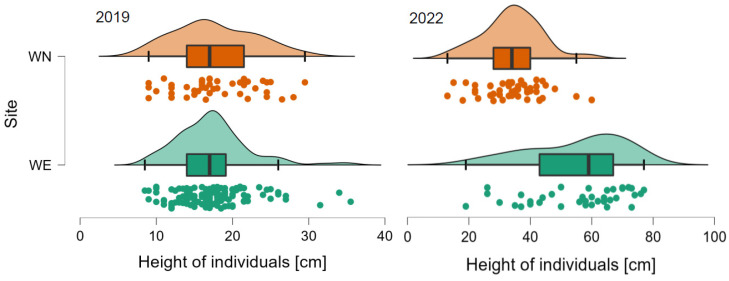
Effect of habitat conditions on the growth rate of *Salix lapponum* individuals. Individuals reintroduced in 2019 on inactive quaking bog in the eastern part of Lake Wiejki (WE) three years later (2022) were almost twice as tall as those planted on active quaking bog in the northern part of the site (WN).

**Figure 5 ijerph-20-00608-f005:**
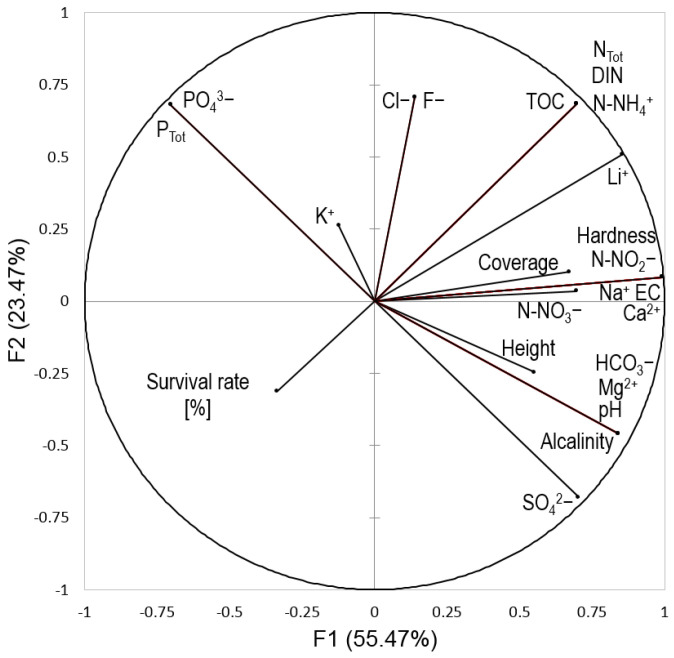
The PCA of the interrelations between the survival rates of *Salix lapponum*, acrotelm water chemistry, and the cover and height of the herbaceous layer.

**Table 1 ijerph-20-00608-t001:** The success of *Salix lapponum* reintroduction performed in 2018–2020 at four sites in the Knyszyn Forest, measured by the number of individuals that survived and produced inflorescences. Seedlings were planted in different habitats in clusters of 24 to 75 individuals.

Location		Aggregation Number	Number of Individuals Introduced	Number of Surviving Individuals	Percentage of Surviving Individuals [%]	Number of Flowering Individuals	Percentage of Flowering Individuals [%]	Time Since Reintroduction [months]
Stare Biele (SB)		1SB	24	20	83.33	0	0	34
		2SB	24	11	45.83	0	0	34
		3SB	24	20	83.33	0	0	34
		4SB	24	15	62.50	0	0	34
		5SB	24	9	37.50	0	0	34
		6SB	24	13	54.17	0	0	34
		7SB	24	10	41.67	0	0	34
		8SB	24	18	75.00	0	0	34
		9SB	24	4	16.67	0	0	34
Derazina valley (DD)		10DD	48	48	100.00	0	0	36
		11DD	24	15	62.50	0	0	36
		12DD	24	19	79.17	0	0	36
Lake Wiejki–east (WE)		1WE	24	0	0	-	-	36
		2WE	24	8	33.33	0	0	36
		3WE	75	29	38.67	9	31.03	36
		4WE	38	36	94.74	0	0	46
		5WE	18	17	94.44	0	0	46
		6WE	33	32	96,97	1	3.13	46
Lake Wiejki–north (WN)		7WN	52	37	71.15	0	0	33
		8WN	51	35	68.63	6	17.14	33
		16WN	52	49	94.23	15	30.61	33
		17WN	51	47	92.16	6	12.77	33
SB total			216	120	55.56	0	0	
DD total			96	82	85.42	0	0	
WE total			212	122	57.55	10	8.20	
WN total			206	168	81.55	27	16.07	

**Table 2 ijerph-20-00608-t002:** Eigenvalues and loading of the calculated principal components.

	F1	F2	F3	F4	F5	F6
**Eigenvalue**	13.313	5.634	3.152	1.440	0.374	0.088
**Variability [%]**	55.471	23.474	13.132	6.000	1.556	0.366
**Cumulative [%]**	55.471	78.945	92.078	98.078	99.634	100.000

## Data Availability

Some or all of the data and models that support the findings of this study are available from the corresponding author upon reasonable request.

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
