# Peer review of "Microhabitat Conditions and Inter-Species Competition Predict the Successful Restoration of Declining Relict Species Populations"

_ijerph, 2022, doi:10.3390/ijerph20010608_

Round 1

Reviewer 1 Report

The article is well prepared, the information is logical and coherent. The topic of conservation and restoration of endangered species is very relevant.  

1. In the introduction, I would suggest that the authors at least briefly discuss the hybridisation of Salix lapponum with other species of the genus, as it has a significant impact on the stability and conservation of the species' populations. 

2. I would suggest that the authors specify more precisely which plant parts were taken for micropropagation, the size to which individuals were grown in artificial media, the size or age at which they were transplanted into the soil, and the size at which they were reintroduced into the wild. It is not enough to say that the individuals were "appropriate size" (line 178 onwards).

3. Line 226. The authors write 'seedlings' but the individuals were produced by micropropagation. Is this a misuse of the term, or were the individuals obtained by growing them from seed? The same in line 408, etc.

4. Table 1 gives the number of individuals that flowered, but it is not clear in which year the planted individuals started flowering. This is an important indicator and should be mentioned in the table or text. 

5. In Fig. 2, I suggest indicating above the boxplot the statistical differences between the study variants.  

6. Have the authors not tried to look at the effect of the location of the tissue used for micropropagation in the plant on the growth and flowering of an individual? It is known that the growth of some conifers is strongly related to the position of the tissue: individuals grown from apical shoots grow rapidly, whereas those from lateral shoots grow much more slowly and are usually stunted. Even if the authors did not take this into account during the study, it could be addressed in the discussion.

7. The occasional spelling and technical errors need to be corrected. 

Reviewer 2 Report

A very valuable paper about a well thought-over and well designed experiment in which a wealth of data have been collected and analysed leading to new insights into succesful practice and recommendations for reintroduction of an endangered willow species and more in general about do's and dont's in reintroduction of shrubs in peatlands. Very valuable study in the sense that the experiment was set up huge with large numbers of individuals and 2-3 years of monitoring. I wonder if in Table 1 an indication can be given about the number of days or the period between reintroduction and the measurement of survival. It is clear that this is between 2 and 3 years, but an indication in for example average number of days +/- SD or so would be helpful. It is not clear to me if these data are available.

Furthermore, I recommend language editing by a native speaker to further imporev the readibility. Text can be more concise, especially in the Discussion section, which is very long and goes into many details and extensive description of potential explanations for the survival rate. shortened.
